# Screening and Characterization of New *Acetobacter* *fabarum* and *Acetobacter* *pasteurianus* Strains with High Ethanol–Thermo Tolerance and the Optimization of Acetic Acid Production

**DOI:** 10.3390/microorganisms10091741

**Published:** 2022-08-29

**Authors:** Taoufik El-Askri, Meriem Yatim, Youness Sehli, Abdelilah Rahou, Abdelhaq Belhaj, Remedios Castro, Enrique Durán-Guerrero, Majida Hafidi, Rachid Zouhair

**Affiliations:** 1Laboratory of Plant Biotechnology and Bio-Resources Valorization, Department of Biology, Faculty of Sciences, Moulay Ismail University, Zitoune, Meknes 50050, Morocco; 2Analytical Chemistry Department, Faculty of Sciences-IVAGRO, Agrifood Campus of International Excellence (CeiA3), University of Cadiz, Polígono Río San Pedro, s/n, 11510 Cadiz, Spain; 3Laboratory of Ecology and Biodiversity of Wetlands Team, Department of Biology, Faculty of Sciences, Moulay Ismail University, Zitoune, Meknes 50050, Morocco

**Keywords:** acetic acid bacteria, *Acetobacter fabarum*, *Acetobacter pasteurianus*, acetic acid tolerant, ethanol–thermo-tolerant strains, pH

## Abstract

The production of vinegar on an industrial scale from different raw materials is subject to constraints, notably the low tolerance of acetic acid bacteria (AAB) to high temperatures and high ethanol concentrations. In this study, we used 25 samples of different fruits from seven Moroccan biotopes with arid and semi-arid environmental conditions as a basic substrate to isolate thermo- and ethanol-tolerant AAB strains. The isolation and morphological, biochemical and metabolic characterization of these bacteria allowed us to isolate a total number of 400 strains with characters similar to AAB, of which six strains (FAGD1, FAGD10, FAGD18 and GCM2, GCM4, GCM15) were found to be mobile and immobile Gram-negative bacteria with ellipsoidal rod-shaped colonies that clustered in pairs and in isolated chains. These strains are capable of producing acetic acid from ethanol, growing on peptone and oxidizing acetate to CO_2_ and H_2_O. Strains FAGD1, FAGD10 and FAGD18 show negative growth on YPG medium containing D-glucose > 30%, while strains GCM2, GCM4 and GCM15 show positive growth. These six strains stand out on CARR indicator medium as isolates of the genus *Acetobacter* ssp. Analysis of 16S rDNA gene sequencing allowed us to differentiate these strains as *Acetobacter fabarum* and *Acetobacter pasteurianus*. The study of the tolerance of these six isolates towards pH showed that most of the six strains are unable to grow at pH 3 and pH 9, with an ideal pH of 5. The behavior of the six strains at different concentrations of ethanol shows an optimal production of acetic acid after incubation at concentrations between 6% and 8% (*v*/*v*) of ethanol. All six strains tolerated an ethanol concentration of 16% (*v*/*v*). The resistance of the strains to acetic acid differs between the species of AAB. The optimum acetic acid production is obtained at a concentration of 1% (*v*/*v*) for the strains of FAGD1, FAGD10 and FAGD18, and 3% (*v*/*v*) for GCM2, GCM4 and GCM15. These strains are able to tolerate an acetic acid concentration of up to 6% (*v*/*v*). The production kinetics of the six strains show the highest levels of growth and acetic acid production at 30 °C. This rate of growth and acetic acid production is high at 35 °C, 37 °C and 40 °C. Above 40 °C, the production of acid is reduced. All six strains continue to produce acetic acid, even at high temperatures up to 48 °C. These strains can be used in the vinegar production industry to minimize the load on cooling systems, especially in countries with high summer temperatures.

## 1. Introduction

Acetic acid (AA) is an organic acid of great global importance. The production of this acid has attracted more and more interest over the years [1]. A total of 75% of this acid is produced synthetically and only 10% of the world production is realized biologically [2,3]. AA production, based on microbial fermentation, is considered a clean and potential alternative for the use of agricultural and biological wastes as carbon sources [4]. Acetic acid bacteria (AAB) are considered a large group of microorganisms that produce AA from ethanol. These bacteria are widely distributed on the surface of flowers and fruits, in sweet substances or in alcoholic beverages [5,6].They are strict aerobic, Gram-negative or variable and catalase-positive microorganisms [7].

The AAB belong to the family *Acetobacteraceae*: they are currently classified in 19 genera (*Acetobacter*, *Acidomonas*, *Ameyamaea*, *Asaia*, *Bombella*, *Commensalibacter*, *Endobacter*, *Gluconacetobacter*, *Gluconobacter*, *Granulibacter*, *Komagataeibacter*, *Kozakia*, *Neoasaia*, *Neokomagataea*, *Nguyenibacter*, *Saccharibacter*, *Swaminathania*, *Swingsia*, and *Tanticharoenia*) [8] and there are 92 species identified to date [9]. The main species most used in vinegar production due to their significant abilities to oxidize ethanol to AA belong to the genera *Acetobacter*, *Gluconacetobacter*, *Gluconobacter* and *Komagataeibacter* [7,10,11,12]. According to these authors, *Acetobacter aceti*, *Acetobacter cerevisiae*, *Acetobacter malorum*, *Acetobacter oeni*, *Acetobacter pasteurianus*, *Acetobacter pomorum*, *Gluconacetobacter entanii*, *Gluconacetobacter liquefaciens*, *Gluconobacter oxydans*, *Komagataeibacter europaeus*, *Komagataeibacter hansenii*, *Komagataeibacter intermedius*, *Komagataeibacter medellinensis*, *Komagataeibacter oboediens* and *Komagataeibacter xylinus*, are the most frequently used species in vinegar production, by a two-step process: alcoholic and acetic fermentation, performed by yeasts and AAB, respectively [2]. Vinegar production is affected by environmental conditions. These limiting factors impact the behavior of AAB by affecting their growth and production capacities through parameters such as temperature, pH, oxygen supply, nutritional inputs (salts, vitamins, glucose, etc.), as well as ethanol concentration in the culture medium [13]. Depending on the intensity of these magnitudes, they can have an influence that aims to slow down or stop the metabolism of the AAB [14]. These inhibitory substances are difficult to avoid since they are often substrates, products of the biochemical reaction, or external conditions.

The optimal temperature of most AAB for AA production is 25 °C to 30 °C [15]. This mesophilic character of AAB, however, presents a considerable disadvantage for industrial applications [6]. The high temperatures during the summer harvest period in some countries, in addition to the heat accumulated during fermentation, are a significant challenge for the vinegar industry [16]. Continuous use of a cooling system is required in this case to maintain optimal temperatures for AAB growth and vinegar production [17]. In the vinegar production industry, temperature control leads to high energy consumption, resulting in increased production costs [18].

In this context, several authors have focused on the isolation and characterization of thermotolerant AAB from tropical products [19,20] that are capable of producing AA at high temperatures, in order to minimize the loads due to the cooling system. The high production of AA at these high temperatures can be explained by a possible temperature tolerance acquired by AAB from their natural zones or from transfer zones [21]. According to [17], *A. aceti* subspecies *aceti* is capable of producing AA at 37 °C after a three-day lag phase of production. Other authors have isolated strains of *A. pasteurianus* from palm wine, Cacao and apple that are capable of producing AA at 39 °C [22], at 47 °C [3] and at 40 °C [23], respectively.

Tolerance in AAB to high temperatures can be achieved by thermal adaptation of the isolated strains. This thermal adaptation plays an important role not only in the reduction of the cooling system, but also in the protection of the fermentation process against accidental failures of the thermal management [24]. Furthermore, thermal adapted strains are very useful in the fermentation process in an acetofermenter at high temperatures, and the use of these strains can decrease the electricity consumption required for cooling by up to 8.5% [25].

In Morocco, studies on thermotolerant AAB have been carried out by [26]. According to these authors, *A. pasteurianus* strains AF01 and CV01, isolated from apple and cactus fruit, produced AA at temperatures up to 41 °C. Similarly, strains isolated from cactus fruit showed acetic acid production at 40 °C [27,28].

The concentration of ethanol in the fermentation medium plays an important role on the membrane permeability and fluidity of AAB. A high concentration leads to a decrease in growth rate, cell viability, metabolic activity and AA production capacity. In general, ethanol tolerance is a species- and strain-dependent trait [29]. Depending on the strains studied, optimal AA production is achieved at concentrations between 4% (*v*/*v*) and 8% (*v*/*v*) of ethanol [23].

During acetic fermentation, the control of the concentration of AA in the medium is essential. When the concentration of AA exceeds 0.5 wt%, the transmembrane proton gradient essential for ATP synthesis is disrupted, causing cellular metabolism to shut down by dissociating AA in microbial cells, leading to an increase in acetate anions and a decrease in intracellular pH [30]. AA tolerance of up to 10% (*v*/*v*) in *K. europaeus* and 6% (*v*/*v*) in *A. pasteurianus* has been recorded [31]. Thus, strains isolated from fermented mango alcohol belonging to the genus *Gluconoacetobacter* represent a 6% (*v*/*v*) resistance to AA [32].

The maintenance of pH is also an essential factor for the survival of AAB. The pH limits reported in the literature for maintaining growth of AAB are within an optimal tolerance range between pH 4.0 and 7.0 [27,33]. In general, most AAB cannot grow at pH levels below 3.0 and above 8.0 [33].

It is within this framework that this work aims to contribute to the isolation of AAB from several substrates scattered in several Moroccan regions with arid and semi-arid environmental conditions. Our concern is to focus on thermo–ethanol-tolerant bacteria, able to produce AA at high temperatures and to resist high concentrations of ethanol and AA. In order to minimize the loads due to the cooling system of the biotechnological processes of vinegar production, the isolated strains were subjected to biochemical and molecular characterization as well as the evaluation of their capacity to produce acetic acid at different stress conditions.

## 2. Materials and Methods

### 2.1. Sample Collection

A total of 25 samples composed of fruits, juices, honeys and vinegars were selected according to their sugar content and their natural habitats. These substrates were collected in seven biotopes of Morocco.

### 2.2. Isolation and Identification of Bacterial Strains

In order to allow the bacteria to regenerate, 5 g or 5 mL of each sample was enriched in a broth enrichment medium consisting of 1.0% glucose, 0.5% (*v*/*v*) ethanol, 0.3% acetic acid, 1.5% peptone and 0.8% yeast extract. The vials were incubated at 30 °C for 7 days [34,35]. After dilution, 0.1 mL(10^−5^ and 10^−6^ dilutions) of aliquot were spread onto a potato agar plate containing: 0.5% glucose, 2% glycerol, 1% yeast extract, 1% peptone, 1.5% potato extract, 4% (*v*/*v*) ethanol, 0.003% bromocresol purple and 2% agar [36], and on a modified CARR Agar medium containing: 3 g/L glucose, 10 g/L CaCO_3_, 0.04 g/L bromothymol blue, 10 g/L yeast extract, 20 g/L agar and 17.5 mL/L ethanol, pH 6.8. Incubation was performed at 30 °C for 48 h [37,38]. Yellowish colonies on potato agar plate and CARR medium were purified on (YPG) medium containing: 0.7% yeast extract, 0.7% peptone, 1% glucose and 2% agar (YPG) medium under aerobic conditions at 37 °C for 48 h.

### 2.3. Morphological Biochemical and Metabolic Tests

Pure bacterial colonies underwent macro and microscopic observations by studying the shape, size, arrangement, Gram staining, pigmentation and motility test, by inoculating the colonies in YPG medium at an incubation temperature of 30 °C for 72 h. Classical biochemical tests, such as catalase activity, cytochrome oxidase, growth in peptone, presence of pigmentation in YPG medium and growth on YPG medium containing D-glucose > 30% were used according to the protocol described by [5,39]. Acetate oxidation and ethanol overoxidation to C_2_O and H_2_O were performed to distinguish between the genera *Acetobacter* and *Gluconobacter*.

### 2.4. Acetic Acid Production Capacity on GYC Medium

According to their morphological and biochemical characteristics, the isolated AAB were evaluated by their ability to produce AA on GYC medium (2% glucose, 0.5% yeast extract, 1.5% calcium carbonate, 1.5% agar, 4% (*v*/*v*) of ethanol, pH 6.8), using the potency index (PI) as a parameter [35]. After incubation at 30 °C for 96 h, the clear zone formed (Figure 1a) in the medium indicated the production of AA, and the size of the clear zone diameters (Figure 1b) revealed the potency of each strain. The diameters of the colonies formed by the isolates and the respective clear zones were measured, and the potency index was determined according to Formula (1). The bacteria with the highest potency index (PI) were selected.
(1)$$\mathrm{Potency}\mathrm{index}(\mathrm{PI})=\frac{\mathrm{Diameter}\mathrm{of}\mathrm{the}\mathrm{formed}\mathrm{clear}\mathrm{zone}\left(\mathrm{mm}\right)}{\mathrm{Diameter}\mathrm{of}\mathrm{the}\mathrm{bacterial}\mathrm{colony}\left(\mathrm{mm}\right)}$$

### 2.5. Tolerance Analysis of AAB Strains

The study of AA production was performed according to the protocol described by [27,40]. Bacterial isolates with the highest potency index (PI) were evaluated by their ability to produce AA on GYC medium at different pH levels (4, 5, 6, 7, 8). To study the AA production capacity of the isolated strains under different stresses, each bacterial suspension was incubated at 30 °C overnight at 120 rpm in GYE medium (3% Glucose, 1.5% Yeast extract, 2.8% (*v*/*v*) Ethanol). Then, 10% of the prepared pre-culture (OD 600 nm = 1.2) was transferred to fermentation medium (2% Glucose, 2% peptone, 7% (*v*/*v*) Ethanol) for 12 days, under agitation at different temperatures (30 °C, 35, 37, 40, 44, 48 °C), at different concentrations of ethanol (4, 6, 8, 12, 14, 16% (*v*/*v*)) and at different concentrations of acetic acid (1, 3, 6% (*w*/*v*)). The bacterial growth rate was monitored and determined every 2 days by measuring the absorbance of the fermentation medium at OD600. The AA content was determined by titration with 0.5 N NaOH, using phenolphthalein as an indicator. The amount of AA in grams produced in 1 L of medium was calculated by the following formula:Acidity (g/L) = [V(NaOH) × 0.5 × 0.06/V (supernatant used)] × 1000 (2)

Three repetitions were realized for each test, and the graphs were realized by GraphPad Prism 8.0.1.

### 2.6. Molecular Identification

Species identification was performed by PCR amplification of the 16S rDNA gene. After culturing of AAB isolates in YPG medium, 1.5 mL of culture was harvested by centrifugation at 12,000× *g* for 2 min and the pellet was washed with 1 mL of sterile distilled water. DNA was extracted using the DNA PureLink^®^ Genomic DNA Mini Kit (Invitrogen, Waltham, MA, USA), and the 16S rDNA region was amplified by using universal primers 27F: (5′-AGAGTTTGATCCTGGCTCAG-3′) and 1492R: (5′-ACGGTTACCTTGTTACGACTT-3′) (Wilson et al., 1990). The PCR reaction mixture (50 μL) contained 43 μL PCR Super Mix and 2.5 μL of each primer. The volume of the mixture was adjusted to 50 μL of sterile distilled water, containing 100 ng of extracted genomic DNA. Each amplification reaction was analyzed on 1% (*w*/*v*) agarose gel in 1 × TBE buffer (pH 8). PCR was performed in a PCR master Cycler, and the reaction parameters were 30 cycles of denaturation at 94 °C for 45 s, annealing at 50 °C for 45 s and extension at 72 °C for 120 s. The amplified products were then purified using the PureLink™ Quick Gel Extraction and PCR Purification Combo Kit (Invitrogen, Waltham, MA, USA) and sequenced in both directions. The obtained sequences were analyzed by individual BLASTn (Basic Local Alignment Search Tool).

Nucleotide sequences were aligned with CLC Genomic workbench 22. Phylogenetic analyses were performed using MEGA version 11 software. The phylogenetic tree was constructed from the alignments by the neighbor-joining method, and the reliability of the inferred trees was tested by the bootstrap test [41].

## 3. Results

### 3.1. Screening and Chemicals Characterization of AAB

From several substrates collected in several Moroccan regions, a total of 400 pure strains of AAB were isolated. These strains are characterized by a yellow coloration on CARR medium characteristics of isolates of the genus *Acetobacter*, and by an acidification capacity with different amplitudes of acid production on GYC medium. Among these isolates, six strains consisting of FAGD1, FAGD10 and FAGD18, isolated from apples, and GCM2, GCM4 and GCM15, isolated from grapes, showed the highest acidification capacity in GYC medium, with a potency index ranging from, 3.6 to 4.0 mm.

Microscopic examination of the 72 h incubation cultures at 30 °C on YPG medium after Gram staining showed Gram-negative bacteria with ellipsoidal rod-shaped colonies that occur in pairs and chains and are non-spore forming. Biochemical tests revealed that all the isolated bacteria were catalase positive, oxidase negative and aerobic obligate. The motility of these six strains was positive in FAGD1, FAGD10 and FAGD18, and negative in GCM2, GCM4 and GCM15. Additionally, the formation of brown pigment on YPG medium was negative, and these isolated strains were also capable of producing AA from ethanol, growing on peptone and oxidizing acetate to CO_2_ and H_2_O under neutral and acidic conditions. Growth at 30% D-glucose concentration differed between the isolated strains: FAGD1, FAGD10 and FAGD18 strains showed negative growth, while GCM2, GCM4 and GCM15 showed positive growth (Table 1). These morphological, cultural and biochemical characters of these isolates show characteristic profiles that correspond to isolates that belong to the genus *Acetobacter*.

### 3.2. Molecular Identification of the Isolates by 16S rDNA Genes Sequencing

Species identification performed by PCR amplification and 16S rDNA gene sequencing showed that isolates FAGD1, FAGD10 and FAGD18 belong to the species *A. fabarum*. The comparison through the phylogenetic tree (Figure 2) constructed with other sequences, obtained with the help of EzTaxom server, showed that the strains FAGD1, FAGD 10 and FAGD18 have an identity of 99.52%, 99.42% and 99.17%, respectively, with *A. fabarum* LMG24244-1 and *A. fabarum* strain OG2 NODE 1lenght 523175-1. The 16S rDNA gene sequencing result of the other three selected isolates, GCM2, GCM4 and GCM15, showed sequence homologies of 99.49%, 97% and 98.47%, respectively, with *A. pasteurianus* strains CICC 22518 and *A. pasteurianus* 386B. The 16S rDNA sequence of these isolated strains have been submitted to NCBI under the accession number ON982715: GCM2, ON982716; GCM4, ON982717; GCM15, ON982718; FAGD1, ON982719; FAGD10, ON982720; FAGD18.

### 3.3. Effect of pH on Acetic Acid Production in Terms of Potency Index (PI)

The tolerance of these six isolates to pH 4 to 8 on GYC medium was studied after incubation of the selected isolates at a temperature of 30 °C for four days. This test showed that the acetic acid production capacity of the isolated strains ranged between pH 4.0 to 8.0, which materialized with PI values varying in a range of 2.7 to 4.26. Most of the six strains were unable to grow at pH 3 and pH 9. The ideal pH for AA production of our six strains, FAGD1, FAGD10, FAGD18, GCM2, GCM4 and GCM15, is pH 5 with potency index (PI) values of 4, 3.93, 3.8, 4.26. 3.76 and 4.2, respectively (Figure 3).

### 3.4. Ethanol Tolerance of AAB Strains

Ethanol in the culture medium is a chemical parameter that can lead to undesirable effects, although it is the substrate for AA fermentation. The rate of AA production of AAB depends on the initial ethanol concentration and careful regulation of this parameter can optimize the AA yield. To investigate the behavior of the six isolated strains towards different ethanol concentrations, we evaluated their growth and AA production after incubation at 30 °C for ten days in ethanol concentrations ranging from 4 to 16% (*v*/*v*) (Figure 4a,b). From this figure, it can be seen that ethanol tolerance is a species- and strain-dependent trait. Optimal AA production was obtained at concentrations of between 6% and 8% (*v*/*v*) ethanol. Increasing the ethanol concentration above these concentrations was progressively accompanied by growth inhibition and a decrease in the acid yield of the six AAB studied. According to Figure 4a,b, the six strains studied can adapt to ethanol concentrations of 14% (*v*/*v*). This ethanol content slightly weakens the growth and AA production of the studied strains, while a content above 16% (*v*/*v*) ethanol stops the growth and AA production for the six strains of *A. fabarum* and *A. pasteurianus*.

### 3.5. Acetic Acid Tolerance of AAB Strains

The optimal growth in the studied strains in the fermentation medium was obtained at a concentration of 3% (*v*/*v*) AA for the strains FAGD1, FAGD10 and FAGD18, and 1% (*v*/*v*) acetic acid for GCM2, GCM4 and GCM15. During the 12 days of fermentation, the production of AA underwent an increase at concentrations of 1% (*v*/*v*) and 3% (*v*/*v*) AA. Meanwhile, at a concentration of 6% (*v*/*v*) AA, a lag phase of 2 to 8 days was recorded according to the strains, followed by a decrease in the amount of AA in the fermentation medium (Figure 5). The recorded lag phase can be explained by an adaptation of the bacteria to this high concentration of AA, while the decrease in the amount of acid in the medium can be explained by a consumption of AA by the strains studied. When the strains were transferred to the same fermentation medium without ethanol and in the presence of the same concentration of AA 6% (*v*/*v*), it was observed that the initial AA level decreased without any lag phase and the growth increased during the 12 days of cultivation (Figure 6). This confirms that, at high concentrations of AA and in the absence of ethanol, the isolated bacteria consume the AA present in the medium to ensure their growth.

### 3.6. Growth and Production of Acetic Acid

The optimum temperature for most AAB for acid production is 25 °C to 30 °C. This mesophilic character of AAB, however, presents a disadvantage during the fermentation process where temperature tolerance is a determining factor for growth rate and AA yield. As the fermentation progresses, a rise in temperature due to rapid heat build-up during the process adds to the optimal temperature of the AAB and becomes a considerable limiting factor for the industrial application. The control of this constraint requires the isolation of thermotolerant AAB, capable of producing AA at high temperatures. In this perspective, and in order to select AAB with high levels of thermotolerance, we evaluated the growth rate and AA yield of six isolates of our two species in fermentation medium. The six isolates were grown in pH 5 medium containing 7% (*v*/*v*) ethanol and at a temperature ranging from 30 °C to 48 °C for a period of twelve days of incubation under agitation. Cell growth and acid yield (Figure 7) were measured every other day. The growth profile of the strains was determined by spectrophotometric analysis at 600 nm, while the AA content was determined by titration with NaOH. The six isolated acetic bacteria strains showed variable growth and production rates in the different incubation temperatures was studied according to the production kinetics. The highest amount of AA production was observed at 30 °C.

Above 37 °C, the exponential phase of acid production was reduced with the increase in the incubation temperature. These six strains showed the highest levels of growth and AA production at 30 °C. The amounts of AA produced at 30 °C reached 49.33 g/L, 53.33 g/L, 47.33 g/L, 53 g/L, 48.66 g/L and 46.33 g/L for strains FAGD 1, FAGD 10, FAGD 18, GCM 2, GCM 4 and GCM 15, respectively. At 35 °C, the amount of AA was between 47 g/L for FAGD 10 and 40.66 g/L for GCM 15, while at 37 °C, an estimated AA production of 40 g/L was obtained by strain FAGD 10. Above this temperature, a strong decrease in AA production was observed for all the strains studied. The results of acetic acid production showed the same trend for the six strains studied: a very low production of AA was observed at 48 °C, with a lag phase of 3 to 6 days.

## 4. Discussion

AAB are widespread microorganisms in nature, on the surface of flowers and fruits, in sweet substances or in alcoholic beverages [5,6]. They are very well known for their ability to oxidize a different range of alcohols and sugars to yield bioacids as end products [42]. In vinegar production, *Acetobacter* species are important and are often used in industrial processes [14]. During the acidification process, the activity of AAB is affected by limiting environmental conditions, such as temperature, pH, dissolved oxygen supply, nutritional inputs (salts, vitamins, glucose, etc.), as well as ethanol concentration in the culture medium [13]. The intensity of these different variables is a major challenge for the vinegar industry. These factors impact the behavior of AAB by affecting their metabolisms through their growth and AA production capabilities [14]. Therefore, the careful choice and selection of AAB that can withstand these extreme stresses is the most important step to optimizing AA production.

Fresh microscopic examination of the six most acidifying strains selected (FAGD1, FAGD10, FAGD18, GCM2, GCM4, GCM15), showed mobile and immobile Gram-negative bacteria, according to the strains studied with ellipsoidal rod-shaped colonies that cluster in pairs, present in isolated chains and do not form spores. These isolated strains are also able to produce AA from ethanol, to grow on peptone and to oxidize acetate to CO_2_ and H_2_O. It should also be noted that strains FAGD1, FAGD10 and FAGD18 showed negative growth on 30% D-glucose while strains GCM2, GCM4 and GCM15 showed positive growth. In addition to their cultural and biochemical profiles, these six strains stood out after 24 h of incubation at 30 °C on CARR medium indicative of acid production by a turn from blue to yellow after 48 h, and then a reversion to blue after a 96 h characteristic of isolates of the genus *Acetobacter ssp* [43].

Cultural and biochemical profiles as methods for phenotypic characterization are not considered reliable enough for bacterial identification. Molecular techniques are the most commonly used methods to identify bacteria [43]. Because of this, 16S rDNA gene sequence analysis was performed to characterize these six selected strains. BLAST analysis of the obtained sequences showed 99.52%, 99.42% and 99.17% sequence identity with *A. fabarum* for strains FAGD1, FAGD10 and FAGD18, and 99.49%, 97% and 98.47% with *A. pasteurianus* for strains GCM2, GCM4 and GCM15.

The study of the tolerance of these six isolates towards a pH ranging from 4 to 8 are unable to grow at pH 3 and pH 9, while the ideal pH for growth is 5. The pH ranges transcribed in the literature for the growth of AAB are within an optimal tolerance range between pH 4.0 and 7.0 [29,33]. In general, most AAB cannot grow at pH below 3.0 and above 8.0 [33]. According to the study conducted by [3], the *A. pasteurianus* strain isolated from apples, and grown in medium containing pure glucose, showed maximum production at pH 5.5, while with pH above 6, AA production was reduced. Other studies have shown growth optimums between pH 5.4 and 6.3 [29]. What is also noteworthy is the maximum growth at pH 8 for an AAB strain, identified as *A. pasteurianus* [23].

The behavior of the six strains after incubation at 30°C for ten days at different ethanol concentrations ranging from 4 to 16% (*v*/*v*), is shown to be species and strain dependent [29]. Indeed, the optimal production of AA is obtained at concentrations between 6% and 8% (*v*/*v*) of ethanol. According to [23], the optimum of AA production was also obtained at ethanol concentrations between 4% and 8% (*v*/*v*). Ethanol is a substrate for acetic acid production but at high concentrations, it inhibits the growth of AAB and limits their acid production. Our work on *A. malorum* strains shows tolerance and/or adaptation to ethanol up to 12%, without profoundly affecting the AA yield [27]. Ethanol tolerance up to 14% (*v*/*v*) was also observed in *A. pasteurianus* strains [3].

The resistance to AA differs among species of AAB. After incubation of isolated strains at different concentrations of AA, the optimum of AA production is obtained at a concentration of 1% (*v*/*v*) for strains of FAGD1, FAGD10 and FAGD18, and 3% (*v*/*v*) in GCM2, GCM4 and GCM15: these strains are able to tolerate a concentration of AA up to 6% (*v*/*v*) with the appearance of a lag phase that differs among strains. Resistance to AA is strongly related to the cell structure and levels of certain enzymes in cell membrane and cytoplasm [9]. During the process of fermentation, *A. pasteurianus* changes their forms, and their behavior differs at different stages with different concentrations of AA [44]. According to [19], this morphological change was directly related to the ability to resist AA, which was accompanied by the formation of a polysaccharide film around the cells of *A. pasteurianus* strains. AAB cells can eliminate the toxic effects of AA on themselves by enhancing enzymatic activities associated with acetic acid uptake, by increasing the capacity of AA transport systems, changing cell morphology and membrane composition and enhancing the expression of molecular chaperones [45]. Analysis of acetic acid-enhanced proteins and overexpression of the gene that encodes this protein [46] endorses that aconitase is involved in AA resistance via enhancement of the TCA cycle. Thus, citrate synthase and aconitase have been considered as major enzymes enabling AA resistance by promoting the TCA cycle process which is related to AA assimilation [44]. According to [30], two groups of mechanisms of tolerance to AA could be applied by bacteria: Mechanisms that aim to reduce the concentration of acetic acid by its catabolism and mechanisms that aim to prevent the entry of acetic acid into the cell and/or suppress the deleterious effects of the internal accumulation of the acid.

The optimum temperature for most AAB for acid production is 25 °C to 30 °C. This mesophilic character of AAB, however, presents a disadvantage during the fermentation process where temperature tolerance is a determining factor for growth rate and AA yield. As the fermentation progresses, a rise in temperature due to rapid heat build-up during the fermentation process becomes a considerable limiting factor for industrial applications. Controlling this constraining factor requires the isolation of thermo-tolerant AAB, which are capable of producing AA at high temperatures to reduce the cooling expense of the acetic fermentation system [3]. In our study, the growth and AA yield profiles of these six isolates show variable growth and production rates. The production kinetics of the six strains show the highest levels of growth and AA production at 30 °C: this rate of growth and AA production decrease at temperatures of 35 °C, 37 °C and 40 °C. Above 40 °C, the exponential phase of acid production is reduced with the increase in the incubation temperature. The same tendency is also observed for the six isolated strains, as they continue to produce AA even at high temperatures up to 48 °C, with a lag phase of 3 to 6 days. Several authors have been interested in the isolation and characterization of thermotolerant AAB from tropical products [19,20] capable of producing AA at high temperatures. Our results are comparable to those obtained on two thermotolerant bacteria designated as *A. tropicalis* and *A. pasteurianus*, isolated from fruits in sub-Saharan Africa. These bacteria also grew well at 35 °C and showed normal growth at 40 and 45 °C [14]. Other authors have isolated strains of *A. pasteurianus* capable of producing AA at 40 °C [3] and 47 °C [23], as well as strains belonging to the genus *Gluconoacetobacter* that are capable of producing AA at a temperature of 45 °C [32]. AAB can adapt to stressful conditions through temporal or permanent acclimation. According to [47], *A. pasteurianus* could be adapted to become a thermo-tolerant bacterium by repeated cultivation at a non-viable temperature and become a thermo-tolerant strain, capable of growing at high temperatures. During industrial fermentation, it is essential to maintain the optimal temperature of the bacterial activity by a cooling system in order to protect the cell growth and the AA production from the heat generated by the fermentation. For this purpose, thermotolerant strains are essential for high temperature fermentation at low cost [47].

## 5. Conclusions

This study allowed us to isolate 400 strains of AAB from different substrates, of which six strains (FAGD1, FAGD10, FAGD18, GCM2, GCM4 and GCM15) were isolated from apple and grape. The morphological, biochemical and molecular characteristics, as well as the properties of these strains to resist different stress conditions, showed that these isolates can produce AA at high ethanol concentrations of 16% (*v*/*v*) ethanol, and can tolerate AA concentration in the fermentation medium up to 6% (*v*/*v*). These thermotolerant strains can produce AA at temperatures up to 48 °C. These strains can be used in the vinegar production industry to minimize the loads due to cooling systems especially in countries with a summer temperature higher than 35 °C, such as Morocco. Nevertheless, additional studies must be carried out to validate other parameters, such as dissolved oxygen and nutritional inputs (salts, vitamins, glucose, etc.).

## Figures and Tables

**Figure 1 microorganisms-10-01741-f001:**
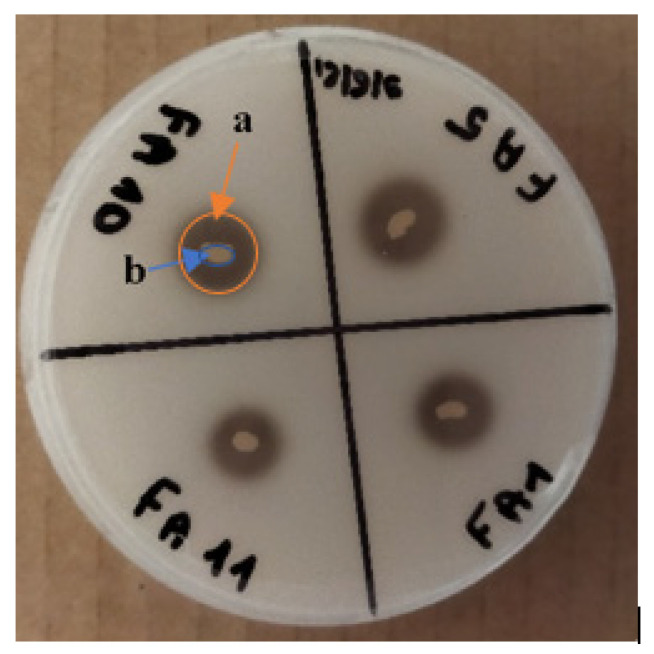
The Acetic Acid Bacteria colonies on GYC media. (a) Clear zone, (b) AAB colony.

**Figure 2 microorganisms-10-01741-f002:**
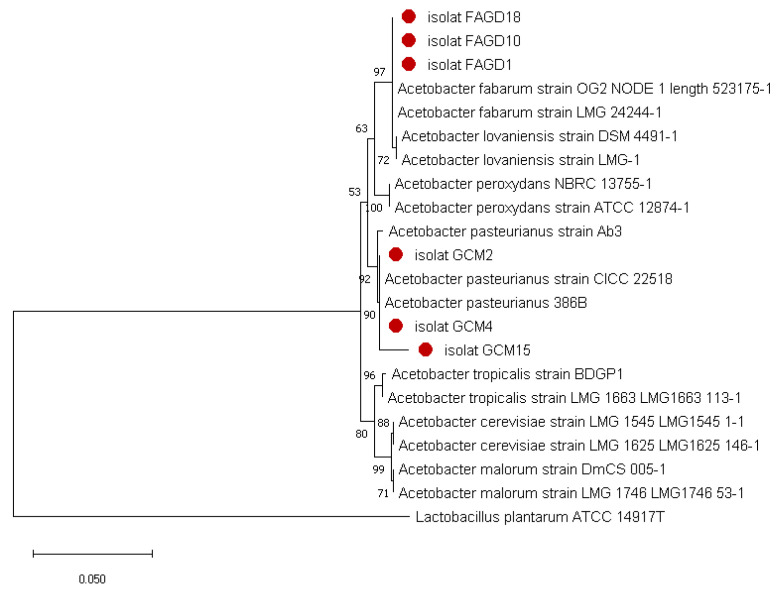
Phylogenetic relationships of the six strains and AAB species based on 16S rRNA gene sequences. *Lactobacillus plantarum* was used as outgroup.

**Figure 3 microorganisms-10-01741-f003:**
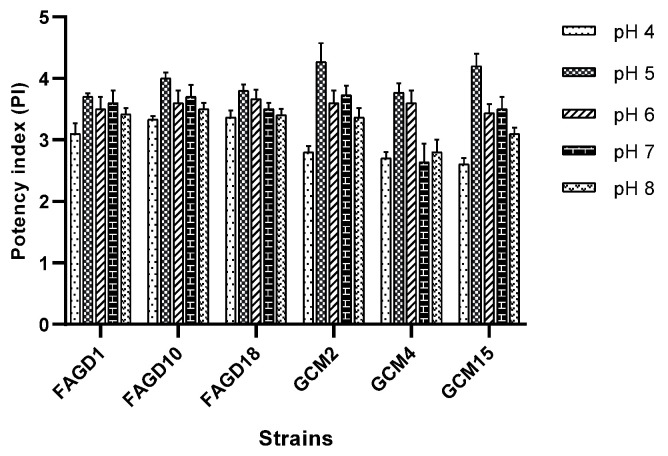
Effect of variation of pH on acidification capacity of six strains.

**Figure 4 microorganisms-10-01741-f004:**
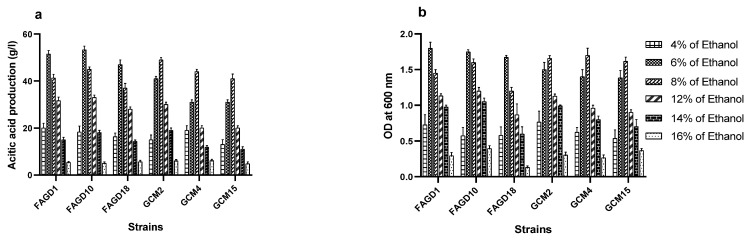
Effect of initial ethanol concentration on acetic acid production (**a**) and bacterial growth (**b**).

**Figure 5 microorganisms-10-01741-f005:**
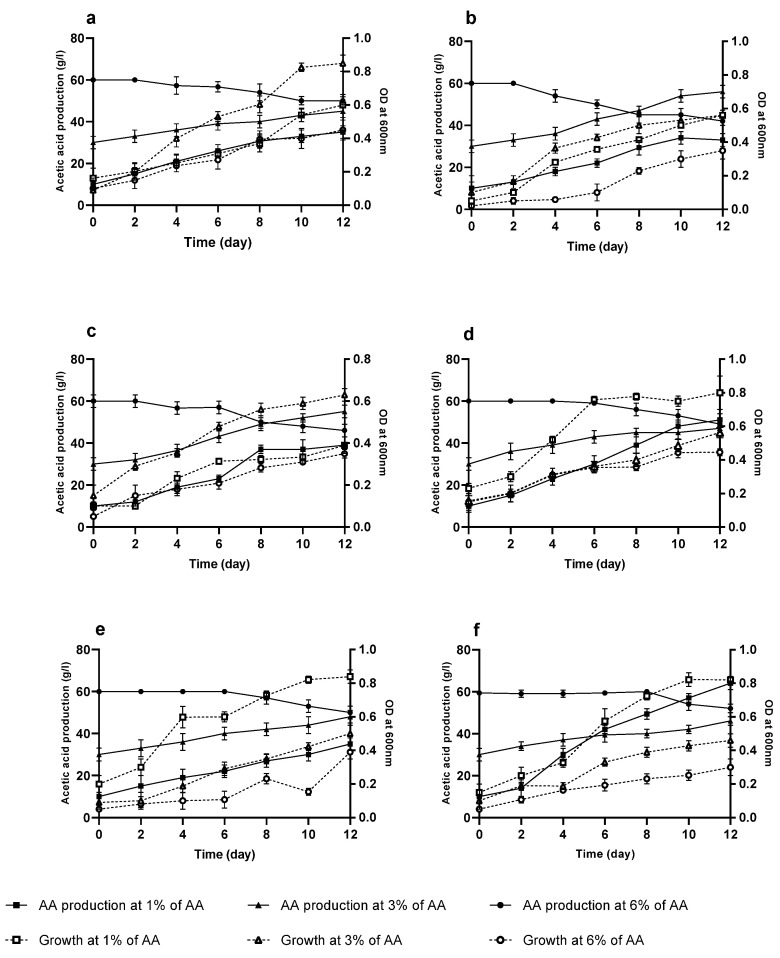
Effect of acetic acid on the production and growth of the six isolated strains. FAGD1 (**a**). FAGD10 (**b**). FAGD18 (**c**). GCM2 (**d**). GCM4 (**e**). GCM15 (**f**).

**Figure 6 microorganisms-10-01741-f006:**
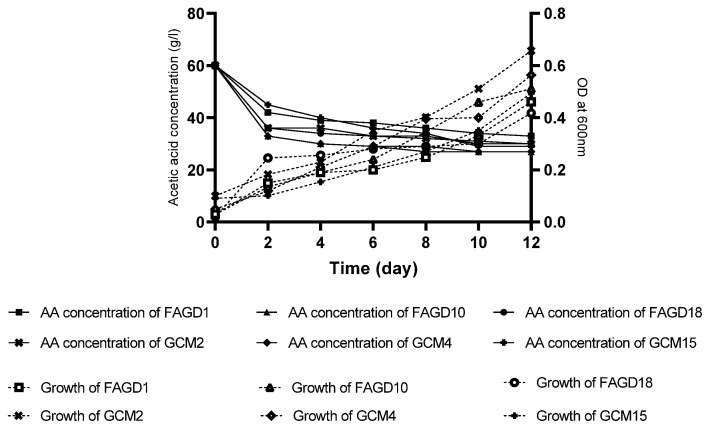
Evolution of acetic acid concentration and bacterial growth at 6% (*v*/*v*) acetic acid.

**Figure 7 microorganisms-10-01741-f007:**
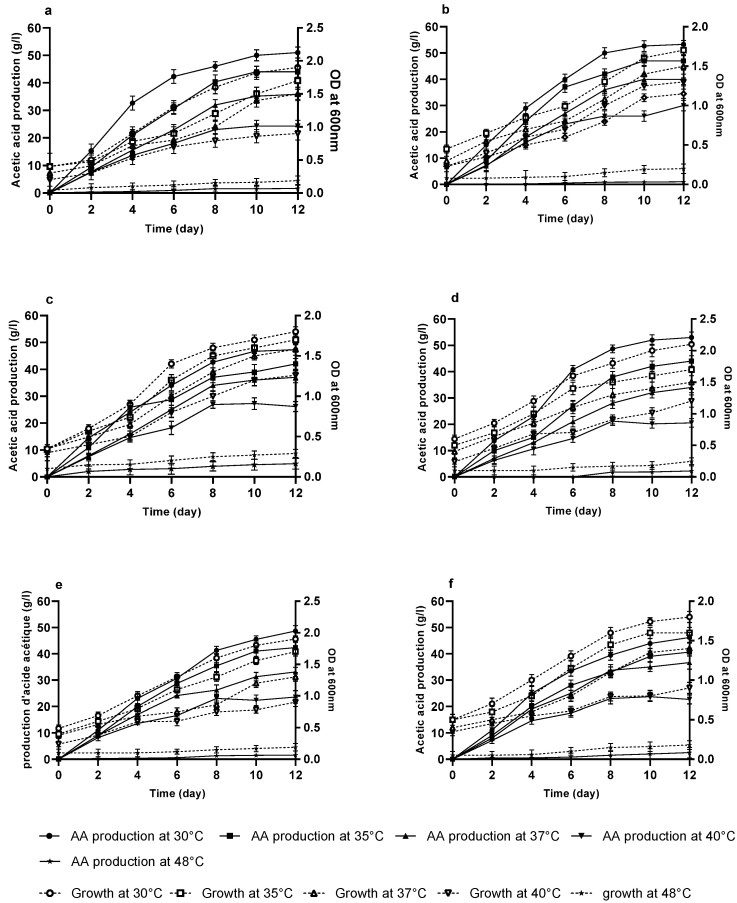
Kinetics of acetic acid production and bacterial growth of the six strains studied: FAGD1 (**a**). FAGD10 (**b**). FAGD18 (**c**). GCM2 (**d**). GCM4 (**e**). GCM15 (**f**).

**Table 1 microorganisms-10-01741-t001:** Morphological and biochemical characteristics of the six selected strains.

Bacterial Isolates							
		FAGD 1	FAGD 10	FAGD 18	GCM 2	GCM 4	GCM 15
Morphological characteristics	Shape	Rod	Rod	Rod	Rod	Rod	Rod
	Gram’s stain	−	−	−	−	−	−
	Spore	−	−	−	−	−	−
	Motility	+	+	+	−	−	−
Biochemical characteristics	Catalase	+	+	+	+	+	+
	Oxidase	−	−	−	−	−	−
	Production of acetic acid from ethanol	+	+	+	+	+	+
	Over oxidation of ethanol to CO_2_ and H_2_O	+	+	+	+	+	+
	Brown pigmentation	−	−	−	−	−	−
	Growth in peptone	+	+	+	+	+	+
	D-glucose concentration > 30%	−	−	−	+	+	+

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
