# Peer review of "Screening and Characterization of New Acetobacter fabarum and Acetobacter pasteurianus Strains with High Ethanol–Thermo Tolerance and the Optimization of Acetic Acid Production"

_microorganisms, 2022, doi:10.3390/microorganisms10091741_

Round 1
Reviewer 1 Report
In this paper, the authors have isolated many acetic acid bacteria (AAB), especially Acetobacter species, from fruits of Morocco, and tried to find out thermo- and ethanol-tolerant AAB strains. For that purpose, several basic studies on strain identification and their characterization on acetic acid fermentation ability, especially related to ethanol tolerance and thermotolerance, have been carried out in this study.
The authors' effort to isolate new and useful AAB strains would be important for development of industrial acetic acid fermentation, and also scientific understanding an evolution and physiology of AAB.
However, as described below, this paper contains several problems from the scientific views and also from the basic knowledges.
1) Basic knowledge on strain description: i) strains' name should be written as italic and starting with Capital letter; Acetobacter but not Acetobacter or acetobacter, ii) both genus name and species name should be written at first appearance but not later; Acetobacter pasteurianus (first) and A. pasteurianus (later). This mistaken description appears every page in the text including References.
2) Several different culture media were used without any description why the medium was used, e.g. YPG (glucose [G], ethanol [E], acetic acid, peptone [P] & yeast extract [YE]), Potato agar plate medium (G, glycerol, YE, P, potato extract, E, bromocresol purple & agar), CAAR medium (G, CaCO3, bromothymol blue, YE, agar & E), GYC solid selective medium (G, YE, CaCO3), GYE medium (G, YE, E), and fermentation medium (G, P, E). In addition, several unclear media also appear: a liquid medium (p. 4, line 3 from the bottom), GYC agar containing CaCO3 (p. 5, line 20), GYP medium (p. 5, line 27), peptone culture medium (p. 5, line 28), GYC medium (p. 7, line 3). Furthermore, "fermentation medium" would be missed in several places (p. 8, lines 6, 14, and p. 10, line10). In case of CAAR medium, bromothymol blue color change may be disturbed with CaCO3?
3) Blast search of 16S rDNA: the authors described "a similarity of 100%" (p. 6, lines 3 and 1 from the bottom) and also "more then 100% sequence similarity" (??, p. 12, line 8 to 7 from the bottom), at the same time, the authors described "formed a separate clade with different strains of A. fabarum" (p. 12, line 4 from the bottom) or "a separate clade with A. pasteurianus" (p. 12, line 3 from the bottom). Actually, based on Fig. 1, the latter description is right but the former descriptions are wrong.
4) Effect of pH on acetic acid production (or growth): The authors described "ideal pH for the growth" based on PI values (p. 7, line 3 to 1 from the bottom), which may be acid production indicator but not colony size. While described "on acetic acid production", only gluconic acid but not acetic acid may be produced in GYC medium (G, YE, CaCO3, p. 4, line 15). And also, there is no description how pH was controlled in this medium.
5) Optimal acetic acid production with different ethanol concentrations: The author described " Optimal acetic acid production is obtained at concentrations between 6% and 8% ethanol" (p. 8, line 8). However, the values are peak at 6%, but no data below 5%.
6) Effect of temperatures on acetic acid production: The authors described "a strong increase in acetic acid production was observed for temperatures of 30°C, 35°C and 37°C" (p. 10, line 8 from bottom), and "This rate of growth and acetic acid production continues to increase at temperatures of 35°C, 37°C and 40°C " (p. 14, line 16-17). Furthermore, they did also " these strains continue to produce acetic acid even at high temperatures up to 48°C" (p. 11, line 2), and "they continue to produce acetic acid even at high temperatures up to 48°C" (p. 14, line 20). However, the production was decreased by increasing the temperature from 30˚C to 40˚C, and any production was seen at 48˚C.
6) Discussion includes largely repetition of the results section.
7) Additional remakes.
i) p. 3. just befor Materials and Methods: Why italic?
ii) p. 4, line 1: "g/L"?
iii) p. 4, line 18: "IP"?
iv) p. 4, line 3 from the bottom: "AAb"?
v) p.5 line 2 from the bottom: Table 1 should be Table 2.
vi) p. 5 line 1 from the bottom: "probably correspond to"?
vii) In Table 1: please describe with English.
viii) In Fig. 5: title is something wrong.
ix) p. 10, line 8: "BBA"?
x) p. 12, line 9: "BBAs"?
Reviewer 2 Report
This article described about screening and characterization of thermotolerant Acetobacter strains isolated from Moroccan biotopes with arid and semi-arid environmental conditions. The results and strains are interesting and worth to be published. However, several things are needed to be improved before acceptance for publication.
1. All through the article, the name of bacteria is not precisely expressed, ex. not italicized, or ‘A. Pasterianus’. And the authors used ‘BAA’ where it should be ‘AAB’. Also, H2O and CO2 should be H2O and CO2. In Discussion, ‘Carr medium’ should be ‘CARR medium’.
2. In introduction, the authors should refer several works about thermotolerant AAB done by the group of Yamaguchi University. As a reference, the reviewer indicates one review “Genomic analyses of thermotolerant microorganisms used for high-temperature fermentations”, Bioscience, Biotechnology, and Biochemistry, (2016), Vol. 80, No. 4, 655–668.
3. The authors express the concentrations of ethanol and acetic acid as %. They should indicate (weight/volume) or (volume/volume).
4. In Results at 3.1, in the second paragraph, the sentence ‘Microscopic examination….’ should be divided into two sentences, because the results of ‘non-spore forming, catalase positive, oxidase negative and aerobic obligate’ are not the results by microscopic examination.
5. In Table 1, ‘Overoxidation of ethanol to CO2 and H2O’ and ‘Oxidation of acetate to CO2 and H2O’ are practically the same things and in fact, the results of all strains are both ‘+’, therefore, the authors should show either of them.
6. In the Result 3.5, the authors should indicate in the first sentence that the results were obtained with GYE medium or fermentation medium, otherwise the obtained results looked like with acetic acid as a sole carbon source.
7. In Figure 5, ‘AA production’ is not appropriate, because in this experiment, acetic acid is just consumed, not produced at all. It should be ‘AA concentration’. Therefore, the title of figure 5 also should be changed. And ’12 days of fermentation’ in the explanation of figure 5 in the text should be changed as ’12 days of cultivation’.
8. In Results 3.6 the author claimed acidification kinetics, but the results are not kinetics but only production. The word ‘kinetics’ is not appropriate and should be changed. And in the last part, ‘latency phase’ should be ‘lag phase’. Furthermore, the lag phase is not obvious as 6 days. The authors said ‘3 to 6 days’ in Discussion and it seems to be more appropriate.
9. In the third paragraph in Discussion, ‘more than 100% sequence similarity’ should be ‘100% identity’. And this paragraph should be moved from Discussion to Results 3.2, because it is not discussion but just results.
Reviewer 3 Report
This is well done study. Though it is a fermentation screening and optimization study it is important and provides a foundation for localized vinegar production that could be unique to the local ecology and therefore potentially more sustainable.
The introduction articulates the study well and methods are reasonable.
The results well written and key essence of results are captured well.
The discussion can be improved highlighting the novelty of the study and implications of the study.
Round 2
Reviewer 1 Report
After the revision, this manuscript was certainly improved well, but still include several problems, which may come from the authors' misunderstanding and/or negligence. Thus, this paper could not be accepted in this form.
1) The authors still have some misunderstanding or real mistakes as for the First query (Point 1) of the referee (description of strains). As the referee mentioned previously, when a strain appeared in the paper at first time, full name of the strain (Genus + species: like Acetobacter pasteurianus) should be written, but after that, it should be abbreviated to be like A. pasteurianus. In this case, the species name should be written with small character "pasteurianus" but not "Pasteurianus".
2) As for the culture media (Point 2), this point was largely improved. But, as for the enrichment culture or YPG medium (p. 4, lines 1-2 from the bottom, and p. 5, line 8), the description is still confused.
3) As for the 16S rDNA sequence (Point 3), the authors have still kept the description "100% similarity". The similarity (better use "identity") should not be 100%, otherwise the phylogenetic tree could not be depicted (Fig. 1). It seems that these Acetobacter species would have the sequence identity in between 97.0 to 99.5%.
4) As for the effect of pH on AA production (Point 4), the authors did not response the referee's concern how AA production was detected on GYC medium. This medium seemed to contain only glucose as the carbon source (p. 5, lines 20-21), and thus only gluconic acid may be produced as the acid.
5) As for the effect of temperature (Point 6), the authors certainly improved this description, but it is strange that "highest" amounts were observed at several different temperatures.
6) As for the discussion (Point 7), the authors certainly tried to improve the discussion. However, some portions still have some ambiguity as follows.
i) The paragraph (p. 17, lines 6-21) does not make sense, and thus better to be re-written. And also, reference [45] of Acetobacter tropicalis may be cited mistakenly with the reference [19] of A. pasteurianus.
ii) Part of sentences on the thermotolerant AAB (p. 18, lines 6-13), the authors had better to summarize all the strains' data comparably, instead of tandemly listed. And also, reference [32] seemed to say AA production at 45˚C but not at 50˚C?
iii) p. 17, lines 9-10 from the bottom; "adds to the optimal temperature" also does not make sense.
7) Additional remark; there are several "acetic acid" instead of AA.
